# Hollow-Core Negative Curvature Fiber with High Birefringence for Low Refractive Index Sensing Based on Surface Plasmon Resonance Effect

**DOI:** 10.3390/s20226539

**Published:** 2020-11-16

**Authors:** Shi Qiu, Jinhui Yuan, Xian Zhou, Feng Li, Qiwei Wang, Yuwei Qu, Binbin Yan, Qiang Wu, Kuiru Wang, Xinzhu Sang, Keping Long, Chongxiu Yu

**Affiliations:** 1State Key Laboratory of Information Photonics and Optical Communications, Beijing University of Posts and Telecommunications, Beijing 100876, China; qiushi@bupt.edu.cn (S.Q.); 1753240191@bupt.edu.cn (Q.W.); quyuwei@bupt.edu.cn (Y.Q.); yanbinbin@bupt.edu.cn (B.Y.); krwang@bupt.edu.cn (K.W.); xzsang@bupt.edu.cn (X.S.); cxyu@bupt.edu.cn (C.Y.); 2Research Center for Convergence Networks and Ubiquitous Services, University of Science & Technology Beijing, Beijing 100083, China; zhouxian219@ustb.edu.cn (X.Z.); longkeping@ustb.edu.cn (K.L.); 3Photonics Research Centre, Department of Electronic and Information Engineering, The Hong Kong Polytechnic University, Hung Hom, Hong Kong; enlf@polyu.edu.hk; 4Department of Physics and Electrical Engineering, Northumbria University, Newcastle upon Tyne NE1 8ST, UK

**Keywords:** hollow-core negative curvature fiber, low refractive index sensing, surface plasmon resonance

## Abstract

In this paper, a hollow-core negative curvature fiber (HC-NCF) with high birefringence is proposed for low refractive index (RI) sensing based on surface plasmon resonance effect. In the design, the cladding region of the HC-NCF is composed of only one ring of eight silica tubes, and two of them are selectively filled with the gold wires. The influences of the gold wires-filled HC-NCF structure parameters on the propagation characteristic are investigated by the finite element method. Moreover, the sensing performances in the low RI range of 1.20–1.34 are evaluated by the traditional confinement loss method and novel birefringence analysis method, respectively. The simulation results show that for the confinement loss method, the obtained maximum sensitivity, resolution, and figure of merit of the gold wires-filled HC-NCF-based sensor are −5700 nm/RIU, 2.63 × 10^−5^ RIU, and 317 RIU^−1^, respectively. For the birefringence analysis method, the obtained maximum sensitivity, resolution, and birefringence of the gold wires-filled HC-NCF-based sensor are −6100 nm/RIU, 2.56 × 10^−5^ RIU, and 1.72 × 10^−3^, respectively. It is believed that the proposed gold wires-filled HC-NCF-based low RI sensor has important applications in the fields of biochemistry and medicine.

## 1. Introduction

Hollow-core negative curvature fiber (HC-NCF), whose surface normal direction of the core boundary is opposite to that of the radial unit vector in the cylindrical coordinate system, propagates the light energy in the core region by the anti-resonance effect [1,2]. Compared with hollow-core photonic band-gap fiber (HC-PBGF), the HC-NCF has great advantages, including simple cladding structure, low transmission loss, large guidance bandwidth, so it has gradually become the research hotspot. In recent years, the HC-NCFs have been demonstrated for low loss transmission [3,4], gas detection [5], temperature sensing [6], etc. Surface plasmon resonance (SPR) effect, which is caused by the collective electronic vibration, generally occurs on the interface between the two media with opposite dielectric constants [7,8]. When the phase-matching condition is satisfied at a certain wavelength, the core mode can be completely coupled with the surface plasmon polariton (SPP) mode, and the sharp confinement loss peak will appear at this time [9,10]. In 1999, Slavı’k et al. firstly proposed a gold-based SPR refractive index (RI) sensor, whose sensitivity and resolution can achieve 3300 nm/RIU and 3 × 10^−5^ RIU in the RI range of 1.324 to 1.335, respectively [11]. Since then, the SPR effect has been proved to be very sensitive to the variation in the RI, so some SPR-based RI sensors have been reported [12,13,14,15]. For the traditional SPR-based RI sensors, the detected RIs are usually larger than 1.35. However, for many commonly used analytes such as sevoflurane, halogenated ethers, and so on, their RIs are located in the RI range of lower than 1.35. Therefore, the SPR-based low RI sensing has urgent application requirements in the fields of biochemistry and medicine [16,17].

At present, with the traditional solid-core photonic crystal fiber (PCF), several SPR-based low RI sensors have been proposed. In 2016, Huang et al. reported a D-shape PCF coated with the indium tin oxide for low RI sensing. The simulation results show that the maximum sensitivity and resolution could reach ~6000 nm/RIU and ~1.6 × 10^−7^ RIU, respectively, in the RI range of 1.28–1.34 [18]. In 2017, Liu et al. demonstrated a mid-infrared PCF coated with the gold for low RI sensing, where the average sensitivity and maximum resolution were 5500 nm/RIU and 7.69 × 10^−6^ RIU, respectively, in the RI range of 1.23–1.29 [19]. In 2018, Dash et al. proposed a PCF coated with the aluminum-doped zinc oxide for low RI sensing, where the maximum sensitivity and resolution could achieve 5000 nm/RIU and 2.0 × 10^−5^ RIU, respectively, in the RI range of 1.32–1.34 [20]. In 2019, Haque et al. achieved a microchannel incorporated PCF coated with the gold and titanium dioxide for low RI sensing, where the maximum sensitivity and resolution were 7000 nm/RIU and 1.96 × 10^−6^ RIU, respectively, in the RI range of 1.22–1.34, while, the sensitivities were 1000 nm/RIU in the RI range of 1.22–1.27 [21]. However, in the previous works, the SPR-based low RI sensors combined with the solid-core PCFs have bad linearity, low resolution, and poor figure of merit (FOM).

In this paper, we propose a gold wires-filled HC-NCF with high birefringence for low RI sensing based on the SPR effect. By introducing the silica tubes with different radii in the cladding region of the designed HC-NCF, the non-degenerate *x*-polarized (*x*-pol) and *y*-polarized (*y*-pol) core modes can be completely coupled with the different SPP even supermodes. The sensing performances of the gold wires-filled HC-NCF in the low RI range of 1.20–1.34 are evaluated by the confinement loss method and birefringence analysis method, respectively. The simulation results show that the proposed gold wires-filled HC-NCF-based low RI sensor can simultaneously achieve high sensitivity, good linearity, high resolution, and large FOM.

## 2. Design of the Gold Wires-Filled HC-NCF and Theory

Figure 1a shows the cross-sectional structure of the designed gold wires-filled HC-NCF, where the black, blue, and yellow regions represent for the silica, analyte, and gold wires, respectively. The cladding region is composed of only one ring of eight silica tubes, the birefringence is introduced by using the silica tubes with different radii, and the angle between the adjacent two silica tubes is 45°. In the four corners (45°, 135°, 225°, and 315°), the inner radius and thickness of the silica tubes are set as *r*_1_ and *t*_1_, respectively. The inner radius and thickness of the silica tubes on the left and right (0° and 180°) are set as *r*_2_ and *t*_2_, respectively. The silica tubes at the top and bottom (90° and 270°) are filled with the gold wires, whose inner radius and thickness are set as *r*_3_ and *t*_3_, respectively. The diameter of the HC-NCF is set as *D*, and the RI of the silica is *n*_silica_. The initial structure parameters of the HC-NCF are set as following: *r*_1_ = 6.7 μm, *t*_1_ = 0.5 μm, *r*_2_ = 6.0 μm, *t*_2_ = 0.5 μm, *r*_3_ = 8.25 μm, *t*_3_ = 0.9 μm, and *D* = 62 μm. In this work, the finite element method is used to investigate the propagation characteristic of the gold wires-filled HC-NCF. In the simulation, the perfectly matched layer (PML) with the thickness of 15 μm and RI of *n*_silica_ + 0.03 is added to the outermost layer of the gold wires-filled HC-NCF. In addition, the grid sizes of the silica and gold wires are set as *λ*/6, and the grid sizes of the analyte and PML are set as *λ*/4. Figure 1b,c show the mode field distributions of the *x*-pol and *y*-pol core modes calculated at wavelength 1.40 μm, respectively. It can be seen from Figure 1b,c that the mode field energies can be well confined in the core region of the gold wires-filled HC-NCF.

*n*_silica_ can be obtained from the widely used Sellmeier equation as [22,23]:(1)$$n_{\mathrm{silica}}(\lambda )=\sqrt{1+\frac{0.6961663\lambda^{2}}{\lambda^{2}-{\left(0.00684043\right)}^{2}}+\frac{0.4079426\lambda^{2}}{\lambda^{2}-{\left(0.1162414\right)}^{2}}+\frac{0.897479\lambda^{2}}{\lambda^{2}-{\left(9.896161\right)}^{2}}}，$$
where *λ* is the free-space wavelength in micrometer.

The dielectric constant of the gold can be calculated by the Drude-Lorentz model [24,25]:(2)$$\varepsilon_{m}=\varepsilon_{\infty }-\frac{\omega_{D}^{2}}{\omega (\omega -j\gamma_{D})}-\frac{\Delta \varepsilon \cdot \Omega_{L}^{2}}{(\omega^{2}-\Omega_{L}^{2})-j\Gamma_{L}\omega },$$
where *ε*_∞_ = 5.9673 and Δ*ε* = 1.09 represent for the permittivity of the high frequency and weighting factor, respectively. *ω*, *γ*_D_, and ω_D_ are the angle frequencies of the guided-wave, damping frequency, and plasma frequency, respectively. Ω_L_ and Γ_L_ stand for the frequency and spectral width of the Lorentz oscillator, respectively. In this work, *γ*_D_/2π = 15.92 THz, *ω*_D_/2π = 2113.6 THz, Ω_L_/2π = 650.07 THz, and Γ_L_/2π = 104.86 THz.

The confinement loss α_Loss_ of the core mode as an important parameter can indicate the strength of the SPR effect, which is described as [26,27]:(3)αLoss (dB/cm)=8.686×2πλ×Im(neff)×104,
where Im(*n*_eff_) stands for the imaginary part of the effective RI.

The sensitivity (*S*) and resolution (*R*) of the HC-NCF-based low RI sensor are defined as [28,29]:(4)$$S\;(\mathrm{nm}/\mathrm{RIU})=\frac{\Delta \lambda_{\mathrm{peak}}}{\Delta n_{a}},$$
(5)$$R\;(\mathrm{RIU})=\frac{\Delta n_{a}\cdot \Delta \lambda_{\min}}{\Delta \lambda_{\mathrm{peak}}},$$
where Δ*λ*_peak_ denotes the resonance wavelength shift, and Δ*λ*_min_ stands for the wavelength resolution of the detector and is set as 0.1 nm. Δ*n*_a_ represents for the RI variation of the analytes.

In order to comprehensively evaluate the performance of the HC-NCF-based low RI sensor, the FOM is defined as the ratio of *S* to full-width-half-maximum (FWHM) of the confinement loss spectrum [30,31]:(6)$$\mathrm{FOM}\;({\mathrm{RIU}}^{-1})=\frac{S\;(\mathrm{nm}/\mathrm{RIU})}{\mathrm{FWHM}\;(\mathrm{nm})}.$$

The birefringence (*B*) is defined as [32,33]:(7)B=Re(neffx)−Re(neffy),
where Re($n_{\mathrm{eff}}^{x}$) and Re($n_{\mathrm{eff}}^{y}$) represent for the real part of the effective RIs of the *x* and *y*-polarized core modes, respectively.

## 3. Influences of the Gold Wires-Filled HC-NCF Structure Parameters on the Propagation Characteristic

For the proposed HC-NCF, the two silica tubes along the *y* direction are filled with the gold wires, so the SPP modes in each gold wire-filled silica tube will occur to couple and form the SPP supermodes. Figure 2a,b show the mode field distributions of the first-order *x*-pol SPP even and odd supermodes calculated at wavelength 1.480 μm, respectively. Figure 2c,d show the mode field distributions of the zero-order *y*-pol SPP even and odd supermodes calculated at wavelength 1.505 μm, respectively. Figure 2e shows the effective RI curves of the *x*-pol and *y*-pol core modes, first-order *x*-pol and zero-order *y*-pol SPP even supermodes and the confinement loss spectra of the *x*-pol and *y*-pol core modes when the RI of the filled analyte is 1.20. Because the silica tubes have different radius, the *x*-pol and *y*-pol core modes are non-degenerate. Thus, the resonance wavelengths of the *x*-pol and *y*-pol core mode are separate. When the real parts of the effective RIs of the core modes and SPP supermodes are equal, the phase-matching condition can be satisfied, and the sharp confinement loss peak will emerge. It can be seen from Figure 2e that the complete coupling occurs between the *x*-pol core mode and first-order *x*-pol SPP even supermode at wavelength 1.485 μm, and the *y*-pol core mode and zero-order *y*-pol SPP even supermode at wavelength 1.510 μm. The resonance wavelengths 1.485 and 1.510 μm also correspond to the positions of the confinement loss peaks. The inserts 1 and 2 in Figure 2e show the mode field distributions of the *x*-pol and *y*-pol core modes calculated at the resonance wavelengths 1.485 and 1.510 μm, respectively.

According to Equation (6), in order to obtain a higher FOM, the FWHM of the confinement loss spectrum needs to be reduced. From Figure 2e, the FWHM of the confinement loss spectrum for the *x*-pol core mode is much narrower than that for the *y*-pol core mode. Thus, the FOM of the *x*-pol core mode is much higher than that of the *y*-pol core mode. In the following, we will investigate the propagation characteristic of the *x*-pol core mode of the gold wires-filled HC-NCF when the fiber structure parameters are changed and the RI of the filled analyte is chosen as 1.20 and 1.21, respectively.

Figure 3a,b show the confinement loss spectra of the *x*-pol core mode of the gold wires-filled HC-NCF when *r*_1_, *t*_1_, and the RI of the filled analyte are changed, respectively. From Figure 3a, when *r*_1_ increases from 6.2 to 7.2 μm and the RI of the filled analyte is chosen as 1.20, the confinement loss peak wavelengths are located at 1.490, 1.485, and 1.478 μm, respectively. When *r*_1_ increases from 6.2 to 7.2 μm and the RI of the filled analyte is chosen as 1.21, the confinement loss peak wavelengths are located at 1.467, 1.462, and 1.455 μm, respectively. As *r*_1_ increases, the resonance wavelengths occur to blue-shift, and the confinement loss peak values increase gradually. The main reason is considered as following: the increase of *r*_1_ leads to the decrease of the core region of the gold wires-filled HC-NCF, and the effective RIs of the *x*-pol core mode and first-order *x*-pol SPP even supermode decrease. Moreover, the *x*-pol core mode easily occurs to couple with the first-order *x*-pol SPP even supermode at this time. From Figure 3b, when *t*_1_ increases from 0.3 to 0.7 μm and the RI of the filled analyte is chosen as 1.20, the confinement loss peak wavelengths are located at 1.486, 1.485, and 1.484 μm, respectively. When *t*_1_ increases from 0.3 to 0.7 μm and the RI of the filled analyte is chosen as 1.21, the confinement loss peak wavelengths are located at 1.463, 1.462, and 1.462 μm, respectively. Compared with the results shown in Figure 3a, it is found that the change of *t*_1_ has little influence on the confinement loss spectra because of the little variation in the effective RIs of the *x*-pol core mode and first-order *x*-pol SPP even supermode.

Figure 4a,b show the confinement loss spectra of the *x*-pol core mode of the gold wires-filled HC-NCF when *r*_2_, *t*_2_, and the RI of the filled analyte are changed, respectively. From Figure 4a, when *r*_2_ increases from 5.5 to 6.5 μm and the RI of the filled analyte is chosen as 1.20 and 1.21, respectively, the confinement loss peak wavelengths and values are almost unchanged. From Figure 4b, when *t*_2_ increases from 0.3 to 0.7 μm and the RI of the filled analyte is chosen as 1.20 and 1.21, respectively, the confinement loss peak wavelengths and values still remain almost unchanged. The main reason is considered that the *x*-pol core mode can be well propagated in the low RI core region of the gold wires-filled HC-NCF by the anti-resonance effect, and the changes of *r*_2_ and *t*_2_ have inapparent influences on the effective RIs of the *x*-pol core mode and first-order *x*-pol SPP even supermode and the coupling between the *x*-pol core mode and first-order *x*-pol SPP even supermode. Although the changes of *r*_2_ and *t*_2_ have little influences on the confinement loss spectra, the two silica tubes on the left and right are indispensable since they play important roles in forming the complete negative curvature boundary of the core region, which is very necessary to achieve the anti-resonance condition.

The confinement loss spectra of the *x*-pol core mode of the gold wires-filled HC-NCF are shown in Figure 5a,b when *r*_3_, *t*_3_, and the RI of the filled analyte are changed, respectively. From Figure 5a, when *r*_3_ increases from 7.75 to 8.75 μm and the RI of the filled analyte is chosen as 1.20, the confinement loss peak wavelengths occur to red-shift, and are located at 1.484, 1.485, and 1.486 μm, respectively. The confinement loss peak values gradually decrease. When *r*_3_ increases from 7.75 to 8.75 μm and the RI of the filled analyte is chosen as 1.21, the confinement loss peak wavelengths also occur to red-shift, and are located at 1.461, 1.462, and 1.463 μm, respectively. The confinement loss peak values decrease slightly. The phenomena can be explained as follows. As *r*_3_ increases, the core region of the gold wires-filled HC-NCF decreases, which reduces the effective RI of the *x*-pol core mode. In contrast, the increase of *r*_3_ enhances the effective RI of the first-order *x*-pol SPP even supermode. Thus, the red-shift of the confinement loss peak wavelengths occurs. In addition, the effective RI difference between the *x*-pol core mode and first-order *x*-pol SPP even supermode gradually increases, and the coupling strength between the *x*-pol core mode and first-order *x*-pol SPP even supermode becomes weaker, which makes the confinement loss peak values decrease. From Figure 5b, when *t*_3_ increases from 0.8 to 1.0 μm and the RI of the filled analyte is chosen as 1.20, the confinement loss peak wavelengths are located at 1.342, 1.485, and 1.625 μm, respectively. When *t*_3_ increases from 0.8 to 1.0 μm and the RI of the filled analyte is chosen as 1.21, the confinement loss peak wavelengths are located at 1.323, 1.462, and 1.599 μm, respectively. It is found that the small change of *t*_3_ can cause the remarkable red-shift of the confinement loss peak wavelength and the slight decrease of the confinement loss peak value. We consider the main reason as following: *t*_3_ is regarded as the barrier between the *x*-pol core mode and first-order *x*-pol SPP even supermode. As *t*_3_ increases, the coupling between the two modes becomes more difficult, which results in the decrease of the confinement loss peak value. At the same time, the increase of *t*_3_ also makes the effective RI of the first-order *x*-pol SPP even supermode increase, so the confinement loss peak wavelength occurs to red-shift.

The confinement loss spectra of the *x*-pol core mode of the gold wires-filled HC-NCF are shown in Figure 6 when *D* and the RI of the filled analyte are changed, respectively. It can be seen from Figure 6 that when *D* increases from 62 to 68 μm and the RI of the filled analyte is chosen as 1.20 and 1.21, respectively, the confinement loss peak wavelengths red-shift from 1.485, to 1.490, and to 1.496 μm, and from 1.462, to 1.467, and to 1.473 μm, respectively, and the corresponding confinement loss peak values gradually decrease. The main reason is considered that as *D* increases, the effective RIs of the *x*-pol core mode and first-order *x*-pol SPP even supermode increase, so the confinement loss peak wavelengths occur to red-shift. In addition, because the effective RI increament of the *x*-pol core mode is larger than that of the first-order *x*-pol SPP even supermode, the effective RI difference between the two modes gradually becomes large. In this situation, it is difficult for the *x*-pol core mode to couple with the first-order *x*-pol SPP even supermode, resulting in the decrease of the confinement loss peak values. From Figure 3, Figure 4, Figure 5 and Figure 6, the fiber structure parameters have different influences on the sensing performances. The detailed results are summarized in Table 1.

## 4. Low RI Sensing Performances of the Gold Wires-Filled HC-NCF

Based on the above analyses, the structure parameters of the gold wires-filled HC-NCF, including *r*_1_, *t*_1_, *r*_2_, *t*_2_, *r*_3_, *t*_3_, and *D*, have different influences on the propagation characteristic. The optimized structure parameters are chosen as following: *r*_1_ = 6.7 μm, *t*_1_ = 0.5 μm, *r*_2_ = 6.0 μm, *t*_2_ = 0.5 μm, *r*_3_ = 8.25 μm, *t*_3_ = 1.5 μm, and *D* = 62 μm. In the following, we will investigate the low RI sensing performances of the gold wires-filled HC-NCF.

Figure 7a shows the confinement loss spectra of the *x*-pol core mode of the gold wires-filled HC-NCF when the RI of the filled analyte changes from 1.20 to 1.34. It can be seen from Figure 7a that as the RI of the analyte increases, the confinement loss peak wavelength occurs to blue-shift, and the corresponding peak value gradually increases. The main reason is considered that as the RI of the filled analyte increases, the effective RI of the *x*-pol core mode obviously increases while that of the first-order *x*-pol SPP supermode changes slightly. Thus, the effective RI difference between the two modes gradually decreases, and the coupling between them becomes stronger. Figure 7b shows the variation in the confinement loss peak wavelength and linear fitting. It can be seen from Figure 7b that the linear fitting result is *y*= −4.667*x* + 7.92474, which means that the average sensitivity is −4667 nm/RIU, and *R*^2^ = 0.996, indicating the good linearity for the sensitivity.

Because of the introduced birefringence, the *x*-pol and *y*-pol core modes of the gold wires-filled HC-NCF can occur to couple with the different SPP even supermodes, as shown in Figure 2e. Thus, except for the confinement loss method, the birefringence analysis method can be also used for investigating the low RI sensing performances [32,33]. Figure 8a shows the effective RI and birefringence curves of the *x*-pol and *y*-pol core modes of the gold wires-filled HC-NCF when the initial fiber structure parameters are used and the RI of the filled analyte is chosen as 1.20. In Figure 8a, the peak 1 can be obtained by the subtraction of the effective RIs between A and B, which is resulted from the coupling between the *x*-pol core mode and first-order *x*-pol SPP even supermode. The dip can be obtained by the subtraction of the effective RIs between C and D, which is resulted from the coupling between the *y*-pol core mode and zero-order *y*-pol SPP even supermode. Peak 2 can be obtained by the subtraction of the effective RIs between E and F, and the absolute birefringence value is larger than those of the peak 1 and dip. This is because the effective RI of the *x*-pol core mode obviously increases after coupling with the first-order *x*-pol SPP even supermode. Figure 8b shows the birefringence curves of the *x*-pol and *y*-pol core modes when the RI of the filled analyte changes from 1.20 to 1.34. From Figure 8b, as the RI of the filled analyte increases, the peaks and dips occur to blue-shift, and the absolute birefringence values decrease, which means that it is easier to detect the lower RI. This is because the effective RIs of the *x*-pol and *y*-pol core modes increase as the RI of the filled analyte increases and the birefringence value is smaller at the shorter wavelength. Figure 8c shows the variation in the confinement loss peak wavelength (peak 2) and its linear fitting when the RI of the filled analyte changes from 1.20 to 1.34. The linear fitting result is *y*= −4.833*x* + 1.18706, and *R*^2^ is 0.996. Compared with the results shown in Figure 7b, the average sensitivity is enhanced from 4667 to 4833 nm/RIU.

Next, several important parameters, including sensitivity, resolution, FOM, and maximum birefringence, need to be considered for evaluating the low RI sensing performances of the proposed gold wires-filled HC-NCF. Figure 9a shows the changes of the sensitivity and FOM with the RI of the filled analyte when the confinement loss method is used. It can be seen from Figure 9a that as the RI of the filled analyte increases, the sensitivity shows an increasing trend, and the minimum and maximum values are 3800 and 5700 nm/RIU in the RI range of 1.20–1.21 and 1.33–1.34, respectively. Besides, the FOM also increases as the RI of the filled analyte increases, and the minimum and maximum values are 152 and 317 RIU^−1^ in the RI of 1.20–1.21 and 1.33–1.34, respectively. When the birefringence analysis method is used, the changes of the sensitivity and maximum birefringence with the RI of the filled analyte are shown in Figure 9b. It can be seen from Figure 9b that as the RI of the filled analyte increases, the sensitivity shows an upward trend, and the minimum and maximum values are 3900 and 6100 nm/RIU in the RI range of 1.20–1.21 and 1.33–1.34, respectively. Compared with the results shown in Figure 9a, the minimum and maximum values are enhanced. In contrast, the maximum birefringence shows a decreasing trend. The birefringence of larger than 1 × 10^−3^ can be achieved in the wide RI range of 1.20 to 1.30, and the minimum and maximum values are 7.63 × 10^−4^ and 1.72 × 10^−3^ when the RI is equal to 1.34 and 1.20, respectively. Figure 9c shows the change of the resolution with the RI of the filled analyte when the confinement loss method and birefringence analysis method are used, respectively. It can be seen from Figure 9c that when the two methods are used, both the resolutions show the decreasing trends as the RI of the filled analyte increases. In the considered RI range of 1.20–1.34, the resolutions are larger than 10^−5^. For the confinement loss method, the minimum and maximum resolutions are 1.75 × 10^−5^ and 2.63 × 10^−5^ RIU in the RI range of 1.33–1.34 and 1.20–1.21, respectively. For the birefringence analysis method, the minimum and maximum resolutions are 1.64 × 10^−5^ and 2.56×10^−5^ RIU in the RI range of 1.33–1.34 and 1.20–1.21, respectively. In summary, the proposed gold wires-filled HC-NCF-based low RI sensor simultaneously achieves high sensitivity, good linearity, high resolution, and large FOM. The comparison results of the low RI sensing performances of the proposed gold wires-filled HC-NCF with other works are summarized in Table 2. From Table 2, compared with other works, the proposed gold wires-filled HC-NCF shows good performances for the low RI sensing.

## 5. Fabrication Processes of the Gold Wires-Filled HC-NCF

The proposed gold wires-filled HC-NCF could be fabricated by the combination of the stack and draw technique and pressure-assisted splicing technique [34,35,36,37]. Figure 10 shows the schematic diagram of the possible fabrication process. First, the silica capillaries are stacked to form the designed structure, and several short capillaries are inserted in the core region at the end of the stack to support the silica tubes in the cladding region and prevent the stack from collapsing. Second, the stack are drawn to form the fiber cane which is at the millimeter scale. Third, the HC-NCF is further drawn from the fiber cane through controlling the gas pressure in the core and cladding regions [38]. Fourth, both the ends of the fiber are cut off, and only the desired structure in the middle part is left. Fifth, a span of short gold wire is put into a silica capillary, and a tungsten wire is used to push the gold wire. It is worth noting that the diameter of the gold wire-filled silica capillary should match that of the silica tube of the HC-NCF which needs to be filled. In addition, the end portion of the gold wire-filled silica capillary needs to be kept clean and fresh. Sixth, melt them together to form a gold capillary, adjust the positions of the gold capillary and silica tube, and splice the gold capillary to the silica tube [39]. Seventh, the spliced section is heated to the melting point of the gold wire (~1064 °C), and the high pressure argon gas is used to push the liquid gold into the silica tube. Finally, the above steps are repeated to fill the gold wire into the other silica tube.

## 6. Conclusions

In summary, we propose a gold wires-filled HC-NCF with high birefringence for low RI sensing based on SPR effect. The structure parameters are optimized by investigating their influences on the propagation characteristic of the gold wires-filled HC-NCF. The sensing performances in the low RI range of 1.20–1.34 are evaluated by the traditional confinement loss method and novel birefringence analysis method, respectively. The simulation results show that when the confinement loss method is used, the proposed gold wires-filled HC-NCF-based sensor can simultaneously achieve the maximum sensitivity of −5700 nm/RIU, resolution of 2.63 × 10^−5^ RIU, and FOM of 317 RIU^−1^. When the birefringence analysis method is used, the proposed gold wires-filled HC-NCF-based sensor can simultaneously achieve the maximum sensitivity of −6100 nm/RIU, resolution of 2.56 × 10^−5^ RIU, and maximum birefringence of 1.72 × 10^−3^. It is believed that the proposed HC-NCF-based low RI sensor has important applications in the fields of biochemistry and medicine.

## Figures and Tables

**Figure 1 sensors-20-06539-f001:**
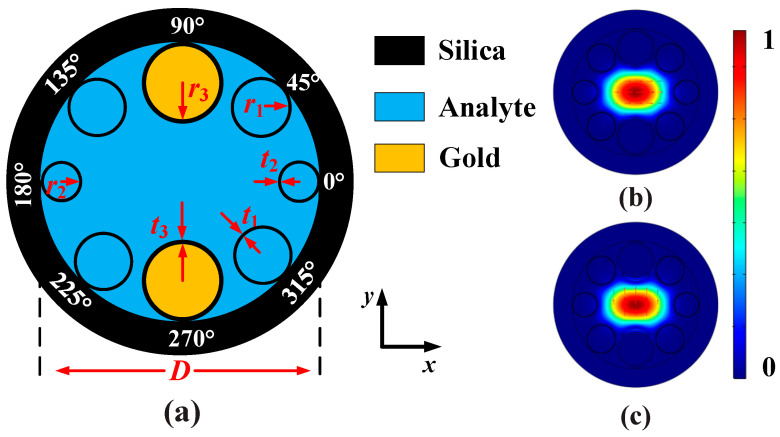
(**a**) The cross-sectional structure of the designed gold wires-filled HC-NCF, where the angle between the adjacent two silica tubes is 45°. (**b**,**c**) show the mode field distributions of the *x*-pol and *y*-pol core modes calculated at wavelength 1.40 μm.

**Figure 2 sensors-20-06539-f002:**
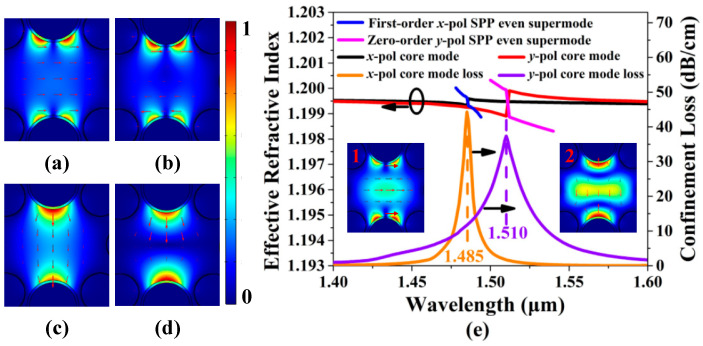
(**a**,**b**) The mode field distributions of the first-order *x*-pol SPP even and odd supermodes calculated at wavelength 1.480 μm, respectively. (**c**,**d**) The mode field distributions of zero-order *y*-pol SPP even and odd supermodes calculated at wavelength 1.505 μm, respectively. (**e**) The effective RI curves of the *x*-pol and *y*-pol core modes, and first-order *x*-pol and zero-order *y*-pol SPP supermodes and the confinement loss spectra of the *x*-pol and *y*-pol core modes when the RI of the filled analyte is 1.20. The inserts 1 and 2 show the mode field distributions of the *x*-pol and *y*-pol core modes calculated at the corresponding resonance wavelengths 1.485 and 1.510 μm, respectively.

**Figure 3 sensors-20-06539-f003:**
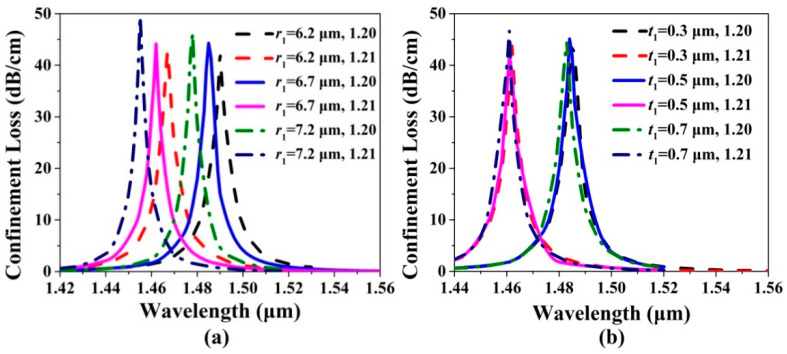
The confinement loss spectra of the *x*-pol core mode when (**a**) *r*_1_ increases from 6.2 to 7.2 μm and (**b**) *t*_1_ increases from 0.3 to 0.7 μm for the RIs of the filled analyte of 1.20 and 1.21, respectively.

**Figure 4 sensors-20-06539-f004:**
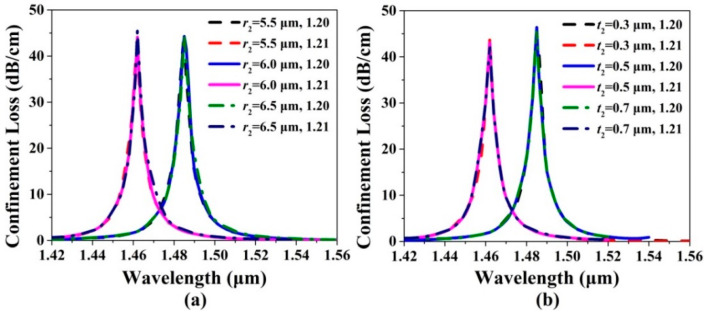
The confinement loss spectra of the *x*-pol core mode when (**a**) *r*_2_ increases from 5.5 to 6.5 μm and (**b**) *t*_2_ increases from 0.3 to 0.7 μm for the RIs of the filled analyte of 1.20 and 1.21, respectively.

**Figure 5 sensors-20-06539-f005:**
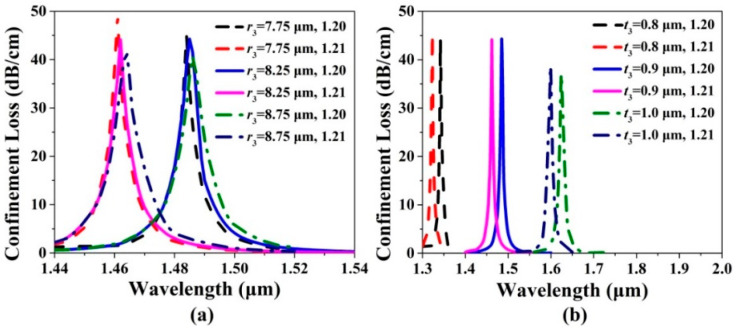
The confinement loss spectra of the *x*-pol core mode when (**a**) *r*_3_ increases from 7.75 to 8.75 μm and (**b**) *t*_3_ increases from 0.8 to 1.0 μm for the RIs of the filled analyte of 1.20 and 1.21, respectively.

**Figure 6 sensors-20-06539-f006:**
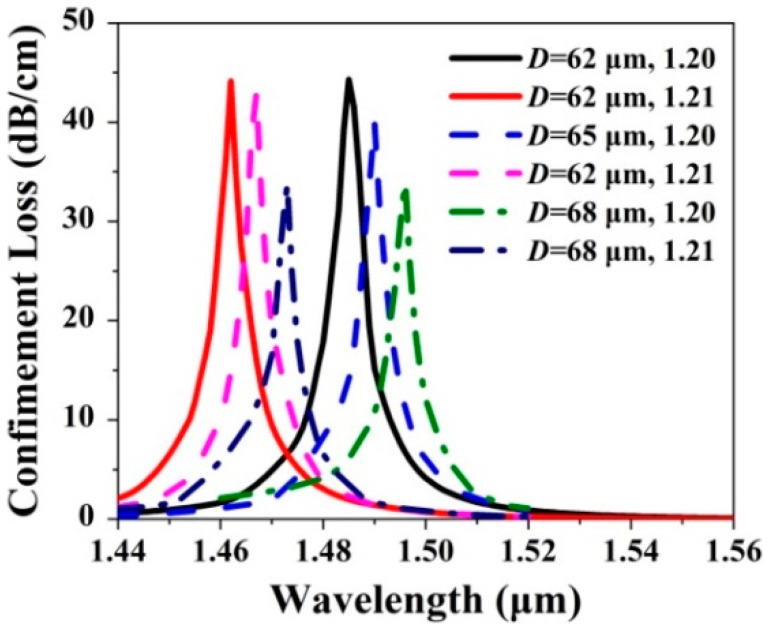
The confinement loss spectra of the *x*-pol core mode when *D* increases from 62 to 68 μm and the RI of the filled analyte is chosen as 1.20 and 1.21, respectively.

**Figure 7 sensors-20-06539-f007:**
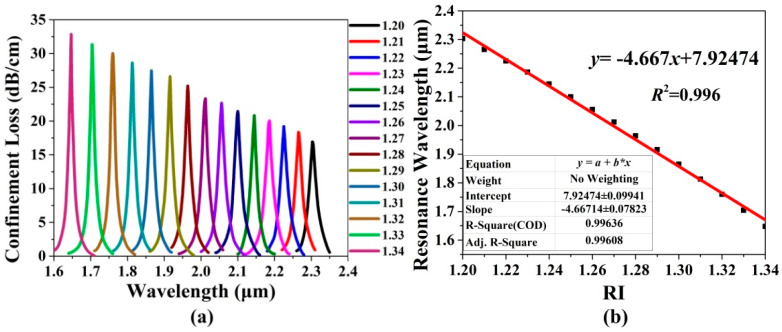
(**a**) The confinement loss spectra of the *x*-pol core mode of the gold wires-filled HC-NCF when the RI of the filled analyte changes from 1.20 to 1.34, and (**b**) the corresponding variation in the confinement loss peak wavelength and linear fitting.

**Figure 8 sensors-20-06539-f008:**
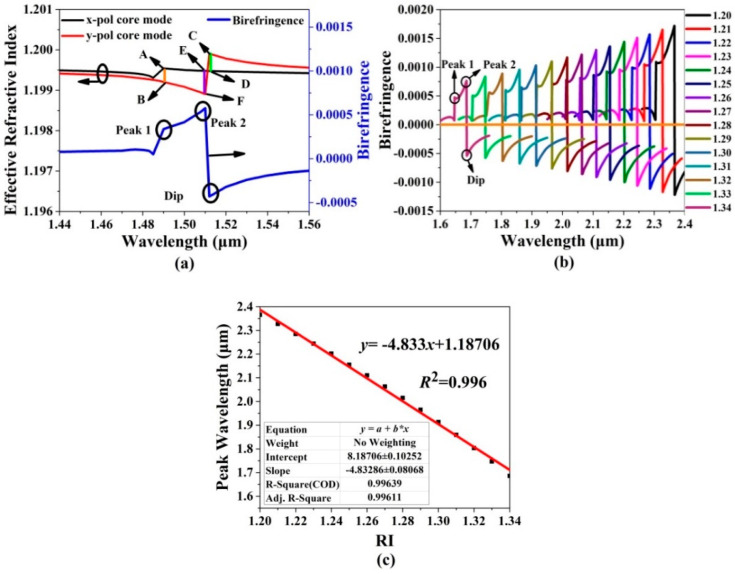
(**a**) The effective RI and birefringence curves of the gold wires-filled HC-NCF when the initial structure parameters of the HC-NCF are set as *r*_1_ = 6.7 μm, *t*_1_ = 0.5 μm, *r*_2_ = 6.0 μm, *t*_2_ = 0.5 μm, *r*_3_ = 8.25 μm, *t*_3_ = 0.9 μm, and *D* = 62 μm, respectively, and the RI of the filled analyte is chosen as 1.20, (**b**) the birefringence curves of the *x*-pol and *y*-pol core modes of the gold wires-filled HC-NCF when the RI of the filled analyte changes from 1.20 to 1.34, and (**c**) the variation in the confinement loss peak wavelength and linear fitting in the considered RI range of 1.20 to 1.34.

**Figure 9 sensors-20-06539-f009:**
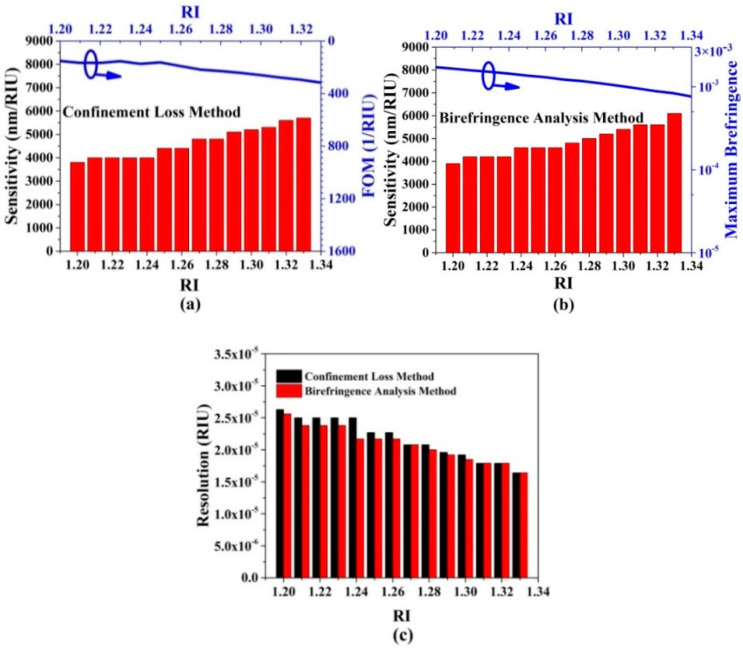
(**a**) The changes of the sensitivity and FOM with the RI of the filled analyte by the confinement loss method, (**b**) the changes of the sensitivity and maximum birefringence with the RI of the filled analyte by the birefringence analysis method, and (**c**) the change of the resolution with the RI of the filled analyte by the confinement loss method and birefringence analysis method, respectively.

**Figure 10 sensors-20-06539-f010:**
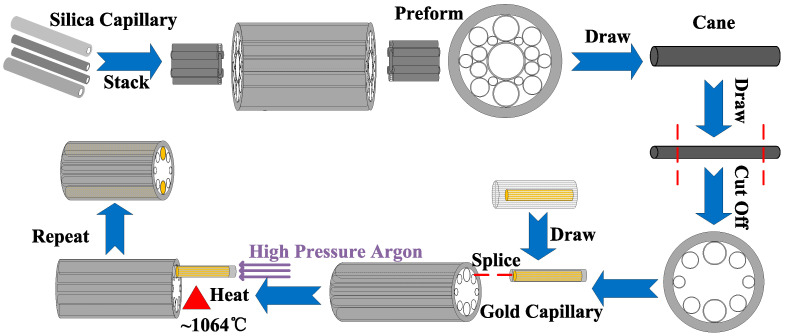
The schematic diagram of the proposed gold wires-filled HC-NCF fabrication by the combination of the stack and draw technique and pressure-assisted splicing technique.

**Table 1 sensors-20-06539-t001:** Summary of the influences of the fiber structure parameters on the sensing performances.

Increase of the Structure Parameters	Resonant Wavelength/Peak Loss	Sensitivity/Resolution	FOM
*r* _1_	Blue-shift/Increased	Unchanged	Slightly Decreased
*t* _1_	Unchanged	Unchanged	Unchanged
*r* _2_	Unchanged	Unchanged	Unchanged
*t* _2_	Unchanged	Unchanged	Unchanged
*r* _3_	Slightly Red-shift/ Decreased	Unchanged	Unchanged
*t* _3_	Red-shift/Decreased	Increased/Decreased	Slightly Decreased
*D*	Red-shift/Decreased	Unchanged	Slightly Decreased

**Table 2 sensors-20-06539-t002:** Comparison results of the low RI sensing performances of the proposed gold wires-filled HC-NCF sensor with other works.

Refs.	Structures	RI Range	|Sensitivity|	Resolution (RIU)	FOM (RIU−1)
[12]	MXene-based SPR RI sensor	1–1.36	N/A (Avg), 198°/RIU (Max)	N/A	N/A
[18]	Indium tin oxide-coated D-shape PCF sensor	1.28–1.34	N/A (Avg), 6000 nm/RIU (Max)	1.6 × 10^−7^	N/A
[19]	Gold-coated SPR PCF sensor	1.23–1.29	N/A (Avg), 5500 nm/RIU (Max)	7.69 × 10^−6^	N/A
[20]	Aluminum-doped zinc oxide-coated PCF sensor	1.32–1.34	N/A (Avg), 5000 nm/RIU (Max)	2.0 × 10^−5^	N/A
[21]	Gold and TiO_2_-coated PCF sensor	1.22–1.34	N/A (Avg), 7000 nm/RIU (Max)	1.96 × 10^−6^	N/A
This work	Gold wires-filled HC-NCF sensor	1.20–1.34	4667 nm/RIU (Avg), 5700 nm/RIU (Max) 4833 nm/RIU (Avg), 6100 nm/RIU (Max)	2.63 × 10^−5^ 2.56 × 10^−5^	317

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
