# Peer review of "Hollow-Core Negative Curvature Fiber with High Birefringence for Low Refractive Index Sensing Based on Surface Plasmon Resonance Effect"

_sensors, 2020, doi:10.3390/s20226539_

Round 1
Reviewer 1 Report
See uploaded file

Author Response
Manuscript Number: Sensors-994510
Title: Hollow-Core Negative Curvature Fiber with High Birefringence for Low Refractive Index Sensing Based on Surface Plasmon Resonance Effect
Authors: Shi Qiu, Jinhui Yuan*, Xian Zhou, Feng Li, Qiwei Wang, Yuwei Qu, Binbin Yan, Qiang Wu*, Kuiru Wang, Xinzhu Sang, Keping Long, And Chongxiu Yu.
Affiliations:
State Key Laboratory of Information Photonics and Optical Communications, Beijing University of Posts and Telecommunications, Beijing 100876, China
Research Center for Convergence Networks and Ubiquitous Services, University of Science & Technology Beijing, Beijing 100083, China
Photonics Research Centre, Department of Electronic and Information Engineering, The Hong Kong Polytechnic University, Hung Hom, Hong Kong
Department of Physics and Electrical Engineering, Northumbria University, Newcastle upon Tyne, NE1 8ST, United Kingdom
Dear Editor,
Thank you very much for your decision about accepting this work after major revisions. We received the reviewers’ comments and would like to thank them for their pertinent and valuable feedback. We made all the necessary changes in the manuscript according to the reviewers’ comments. Please find following detailed descriptions of changes where they are performed. We have copied the reviewers’ comments before responding to it to facilitate the easy reading of this document. The revised parts in the original manuscript are marked in red font.
Please let us know if additional information is necessary to support our manuscript.
Best regards,
Shi Qiu, on behalf of the other authors.
Reviewers' comments:
Reviewer #1: In this manuscript, entitled “Hollow-Core Negative Curvature Fiber with High Birefringence for Low Refractive Index Sensing Based on Surface Plasmon Resonance Effect”, Qiu et al reported the computational design and numerical simulation of a gold-filled hollow-core negative curvature fiber (HCNCF) with high birefringence for low refractive index sensing. The proposed device operates based on the concept of surface plasmon resonance effect, which has also been broadly explored in other materials, such as nanowires and 2D nano thin-films. The Authors have carefully examined how various device parameters could affect the plasmonic resonance effects. It is rigorously demonstrated that the HC-NCF sensor provides a good performance in comparison to many other existing refractive index sensor designs. As refractive index sensor is tremendously useful in biology and chemistry, the computational results presented in this manuscript is important and can potentially generate interests among the sensor and photonics community. The manuscript is thus in good alignment with the scope of Sensors and I recommend the manuscript to be accepted for publication after performing the following minor revisions
1) The minimum wavelength resolution Δλmin in Eq. (5) is an important quantity in determining the resolution (R) of the sensor. It is however not clear what Δλmin is used when calculating R, for example in Fig. 9 and the summarized value in Table 1. The Authors are recommended to briefly describe how Δλmin is determined or estimated in their modeling.
Authors’ response: We thank the reviewer for his/her comments. We have added the explanation of Δλmin in the “2. Design of the Gold Wires-Filled HC-NCF and Theory” section of the revised paper as following
“Δλmin stands for the wavelength resolution of the detector and is set as 0.1 nm.”
In the practical sensing processes, the resolution is not only related to the designed optical fiber, but also to spectrometer. For example, in Eq. (5), Δλpeak and Δna are related to the sensitivity of the designed fiber sensor, and Δλmin is related to the spectrometer. The parameter Δλmin is the smallest wavelength variation that a spectrometer can detect. In our simulation, Δλmin is assumed to be 0.1 nm, as reported in other works [1-3].
[1] Liu. C, Yang. L, Lu. X. Mid-infrared surface plasmon resonance sensor based on photonic crystal fibers. Opt. Express, 2017, 25, 14227-14237.
[2] Hasan M.R, Akter. S, Rifat A.A. Spiral Photonic Crystal Fiber-Based Dual-Polarized Surface Plasmon Resonance Biosensor. IEEE Sens J, 2017, 18,133-140.
[3] Haque E, Hossain M.A, Namihira. Y. Microchannel-based plasmonic refractive index sensor for low refractive index detection. Appl Opt., 2019, 58, 1547-1554.
2) The proposed gold-wire-filled HC-NCF sensor appears to be dependent on a large number of device parameters, such as D, r1, r2, r3, t1, t2, t3 and also the relative orientation of each of the eight tubes inside the larger silica tube. Although detailed numerical simulations are performed and presented in Figs. 3 to 6 to demonstrate how these parameters can affect the confinement loss, a clear summary and/or design guideline and/or parametric optimization of these parameters so to achieve the best sensitivity, resolution and FOM are not provided. The Authors should include a discussion on this aspect so to highlight a design strategy to achieve the highest refractive index sensing capability. Such discussion will greatly improve the experimental relevance and the novelty of this manuscript.
In relevance to this, are the S, R and FOM of the proposed HC-NCF sensor quoted in Table 1 obtained after performing a parametric optimization that scans through all device parameter space – or is it simply a “good configuration” that yields “good value” rather than optimal values?
Authors’ response: We thank the reviewer for his/her comments. We have added the content in “3. Influences of the Gold Wires-Filled HC-NCF Structure Parameters on the Propagation Characteristic” section of the revised paper as following
“From Figs.3 to 6, the fiber structure parameters have different influences on the sensing performances. The detailed results are summarized in Table 1.”
Table 1. Summary of the influences of the fiber structure parameters on the sensing performances
|
Increase of the Structure Parameters |
Resonant Wavelength/Peak Loss |
Sensitivity/Resolution |
FOM |
|
|
r1 |
Blue-shift/Increased |
Unchanged |
Slightly Decreased |
|
|
t1 |
Unchanged |
Unchanged |
Unchanged |
|
|
r2 |
Unchanged |
Unchanged |
Unchanged |
|
|
t2 |
Unchanged |
Unchanged |
Unchanged |
|
|
r3 |
Slightly Red-shift/Decreased |
Unchanged |
Unchanged |
|
|
t3 |
Red-shift/Decreased |
Increased/Decreased |
Slightly Decreased |
|
|
D |
Red-shift/Decreased |
Unchanged |
Slightly Decreased |
|
After analyzing the influences of the fiber structure parameters on the sensing performances, the optimized structure parameters are given in the “4. Low RI Sensing Performances of the Gold Wires-Filled HC-NCF” section as following
“The optimized structure parameters are chosen as following: r1=6.7 μm, t1=0.5 μm, r2=6.0 μm, t2=0.5 μm, r3=8.25 μm, t3 = 1.5 μm, and D=62 μm.”
3) How is the fault tolerance of their design? Particularly, if the radius of the 8 inner tubes fluctuate by, say 1% to 5% which is very common in realistic device fabrication, how will such variation affect the performance in terms of refractive index sensing? The Authors should include some additional simulation to investigate the robustness of the device performance when the various radii, tube thickness and the angular position of each tubes are perturbed. Such “fault tolerance test” could significantly improve the practicality and feasibility of the proposed sensing device.
Authors’ response: We thank the reviewer for his/her comments. We found that increasing of r1 and D can only lead to the peak wavelength shift and the confinement loss peak values changes. While their contributions to the sensitivity, resolution and FOM are little. The small confinement loss peak value differences caused by the changes of r1 and D are not important for the current detection technology. Only the change of t3 has obvious effect on the sensitivity. Meanwhile, as seen from Fig. 3(a) to Fig. 5 (a) in the manuscript, even if the changes of t1, r2, t2 and r3 are larger than 5%, their confinement loss spectra are still coincident, which mean that these structure parameters are insensitive to the sensing performances. So, we study the influences of structural parameters r1, D, and t3 with 2% change on the confinement loss spectra based on the final structure parameters with the RI of 1.20.
Fig.1 shows the fabrication tolerance of the x-pol core mode when (a) r1, (b) D and (c) t3 have the changes of ±2% in final structure parameters, respectively. As seen in Fig. 1(a), when r1 increases the confinement loss peak wavelengths blue shifts by 2 nm each time and the corresponding confinement loss peak values gradually increase. From Fig. 1(b), when D increases the confinement loss peak wavelengths red shifts by 1 nm each time and the corresponding confinement loss peak values gradually decrease. From Fig. 1(c), when t3 increases, the confinement loss peak wavelengths red shifts by 40 nm each time and the corresponding confinement loss peak values gradually decrease.
Fig. 1. The fabrication tolerance of the x-pol core mode when (a) r1, (b) D, and (c) t3 change, respectively.
4) I recommend the Authors to also include a brief comparison with SPR refractive index sensor based on novel 2D materials, such as MXenes and two-dimensional transition metal dichalcogenides [e.g. Xu et al, Nanomaterials 9, 165 (2019)], and graphene [Lin et al, Opt. Expr. 18, 14395 (2010)]. Such comparison could be included in Table 1 and/or in a relevant paragraph, so to provide a broader scope view on the sensing performance of the device proposed in this manuscript.
Authors’ response: We thank the reviewer for his/her comments. We have cited the two important papers as Refs. [11] and [12]. The Ref. [11] has been described in the “1. Introduction” section and the Ref.[12] has been added in the Table 2 of the revised paper as following
“In 2010, Wu et al presented a graphene-on-gold based SPR biosensor, which used the attenuated total reflection method to detect the refractive index (RI) change on the graphene-on-gold surface and significantly improved the detection sensitivity [11]. Since then, the SPR effect has been proved to be very sensitive to the variation in the RI, so some SPR-based RI sensors have been reported [12-15].”
“[11] Wu L., Chu H. S., Koh W. S., Li E.P. Highly sensitive graphene biosensors based on surface plasmon resonance. Opt. Express, 2010, 18, 14395-14400.”
“[12] Xu Y., Ang Y.S, Wu L., Ang L.K. High Sensitivity Surface Plasmon Resonance Sensor Based on Two-Dimensional MXene and Transition Metal Dichalcogenide: A Theoretical Study. Nanomaterials, 2019, 9, 165.”
Table 2. Comparison results of the low RI sensing performances of the proposed gold wires-filled HC-NCF with other works.
|
Refs |
Structures |
RI Range |
|Sensitivity| |
Resolution (RIU) |
FOM (RIU-1) |
|
[12] |
MXene-based SPR RI sensor |
1-1.36 |
N/A (Avg), 198°/RIU (Max) |
N/A |
N/A |
|
[18] |
Indium tin oxide-coated D-shape PCF sensor |
1.28-1.34 |
N/A (Avg), 6000 nm/RIU (Max) |
1.6×10−7 |
N/A |
|
[19] |
Gold-coated mid-infrared SPR PCF sensor |
1.23-1.29 |
N/A (Avg), 5500 nm/RIU (Max) |
7.69×10−6 |
N/A |
|
[20] |
Aluminum-doped zinc oxide-coated PCF sensor |
1.32-1.34 |
N/A (Avg), 5000 nm/RIU (Max) |
2.0×10−5 |
N/A |
|
[21] |
Gold and TiO2-coated PCF sensor |
1.22-1.34 |
N/A (Avg), 7000 nm/RIU (Max) |
1.96×10−6 |
N/A |
|
This work |
Gold wires-filled HC-NCF sensor |
1.20-1.34 |
4667 nm/RIU (Avg), 5700 nm/RIU (Max) 4833 nm/RIU (Avg), 6100 nm/RIU (Max) |
2.63×10-5 2.56×10-5 |
317 |

Reviewer 2 Report
Type of manuscript: Article
Title: Hollow-Core Negative Curvature Fiber with High Birefringence for Low
Refractive Index Sensing Based on Surface Plasmon Resonance Effect
Journal: Sensors
Review comments
This manuscript reported the theoretical investigation of a gold wires-filled HC-NCF proposed for low index sensing based on SPR effects. The study employed the finite element method to calculate the light propagation properties and covered the sensing characteristics in the low index range using confinement loss method and birefringence method. The manuscript has been organized well and delivered interesting points in the optical biosensing fields. However, prior to the publication of this manuscript in the journal, authors need to provide more lines and data for clearer explanation for the concerns as seen in the following:
- Authors claimed that SPR effects occurred in this gold-wired filled structure at wavelength above the telecom wavelength. Does the gold dielectric constant support the SPR at these wavelengths? Authors need to present the complex values of the relative permittivity of gold used in their computation.
- Analyte with its index of 1.20-1.34 would cover liquid phased analyte. Usually, this sort of analyte might absorb light at these wavelengths. Why did authors choose these wavelengths for sensing? If absorption may occur, how can this effect be dealt with?
- As claimed in the Conclusion section, this type of optical sensor could be used for application in the biochemistry and medicine. Authors had better offer a couple of lines on how the biochemical surface treatment required for biochemistry sensing can be performed practically, for example, on which surface of this fiber structure bio-recognition elements such as antibody or aptamer, or DNA can be immobilized.
Author Response
Manuscript Number: Sensors-994510
Title: Hollow-Core Negative Curvature Fiber with High Birefringence for Low Refractive Index Sensing Based on Surface Plasmon Resonance Effect
Authors: Shi Qiu, Jinhui Yuan*, Xian Zhou, Feng Li, Qiwei Wang, Yuwei Qu, Binbin Yan, Qiang Wu*, Kuiru Wang, Xinzhu Sang, Keping Long, And Chongxiu Yu.
Affiliations:
State Key Laboratory of Information Photonics and Optical Communications, Beijing University of Posts and Telecommunications, Beijing 100876, China
Research Center for Convergence Networks and Ubiquitous Services, University of Science & Technology Beijing, Beijing 100083, China
Photonics Research Centre, Department of Electronic and Information Engineering, The Hong Kong Polytechnic University, Hung Hom, Hong Kong
Department of Physics and Electrical Engineering, Northumbria University, Newcastle upon Tyne, NE1 8ST, United Kingdom
Dear Editor,
Thank you very much for your decision about accepting this work after major revisions. We received the reviewers’ comments and would like to thank them for their pertinent and valuable feedback. We made all the necessary changes in the manuscript according to the reviewers’ comments. Please find following detailed descriptions of changes where they are performed. We have copied the reviewers’ comments before responding to it to facilitate the easy reading of this document. The revised parts in the original manuscript are marked in red font.
Please let us know if additional information is necessary to support our manuscript.
Best regards,
Shi Qiu, on behalf of the other authors.
Reviewers' comments:
Reviewer #2: This manuscript reported the theoretical investigation of a gold wires-filled HC-NCF proposed for low index sensing based on SPR effects. The study employed the finite element method to calculate the light propagation properties and covered the sensing characteristics in the low index range using confinement loss method and birefringence method. The manuscript has been organized well and delivered interesting points in the optical biosensing fields. However, prior to the publication of this manuscript in the journal, authors need to provide more lines and data for clearer explanation for the concerns as seen in the following:
1) Authors claimed that SPR effects occurred in this gold-wired filled structure at wavelength above the telecom wavelength. Does the gold dielectric constant support the SPR at these wavelengths? Authors need to present the complex values of the relative permittivity of gold used in their computation
Authors’ response: We thank the reviewer for his/her comments. Many works about the gold-based SPR sensors or devices have been reported. Among of them, the wavelength ranges used are similar with our work, and some of them even extend to the mid-infrared spectral region. Please see the Refs. [1-3] as follows
In 2010, DiPippo et al reported a doped-silicon gold-based SPR sensor in mid-infrared spectral range. The research results indicated that the proposed structure can excite the SPR on the normal incidence of mid-IR light, resulting in a large probing depth. The operation wavelength range is from 4.7 to 6.0 μm.
In 2017, Liu et al demonstrated a mid-infrared PCF coated with the gold for low RI sensing, where the RI detection range is from 1.23 to 1.29 and the operating wavelength range is from 2500 to 3000 nm.
In 2018, Yang et al reported a temperature sensor based on a gold-coated hollow-core fiber (HC-1550-02, produced by NKT Photonics). The operating wavelength is from 1200 to 2400 nm.
[1] Dipippo W, Lee B J, Park K. Design analysis of doped-silicon surface plasmon resonance immunosensors in mid-infrared range. Opt. Express, 2010, 18, 19396-406.
[2] Liu, C., Yang, L., Lu, X., Liu, Q., Wang, F., Lv, J., Sun, T., Mu, H., Chu, P.K., Mid-infrared surface plasmon resonance sensor based on photonic crystal fibers. Opt. Express. 2017, 25 14227–14237.
[3] Yang X., Lu Y., Liu B., et al. High Sensitivity Hollow Fiber Temperature Sensor Based on Surface Plasmon Resonance and Liquid Filling. IEEE Photonics J, 2018, 10, 1-9.
In this paper, the dielectric constant of the gold is calculated by the Drude-Lorentz model as following
where ε∞=5.9673 and Δε=1.09 represent for the permittivity of the high frequency and weighting factor, respectively. ω, γD, and ωD are the angle frequencies of the guided-wave, damping frequency, and plasma frequency, respectively. ΩL and ΓL stand for the frequency and spectral width of the Lorentz oscillator, respectively. In this work, γD/2π=15.92 THz, ωD/2π=2113.6 THz, ΩL/2π=650.07 THz, and ΓL/2π=104.86 THz. The complex values of the relative permittivity of gold in the wavelength range of 1.50 to 2.40 μm are shown as follows:
The complex values of the relative permittivity of the gold
|
λ (μm) |
Relative permittivity |
λ (μm) |
Relative permittivity |
|
1.5 |
-103.96469792+8.91799857i |
2.0 |
-189.48747623+20.92585933i |
|
1.6 |
-119.18097499+10.79414578i |
2.1 |
-209.39766471+24.18698745i |
|
1.7 |
-135.34631143+12.91862983i |
2.2 |
-230.22919934+27.76764414i |
|
1.8 |
-152.45602349+15.30586036i |
2.3 |
-251.97545126+31.68134301i |
|
1.9 |
-170.50494134+17.97018882i |
2.4 |
-274.62948729+35.94202977i |
2) Analyte with its index of 1.20-1.34 would cover liquid phased analyte. Usually, this sort of analyte might absorb light at these wavelengths. Why did authors choose these wavelengths for sensing? If absorption may occur, how can this effect be dealt with?
Authors’ response: We thank the reviewer for his/her comments. The reason we choose the sensing wavelength range is as following. First, the fiber material is silica. When the current silica purification technology and fabrication method are used, the silica absorption loss is relatively low in the considered wavelength range. Second, most of the commercial light sources and spectrometers can match the wavelength range. Third, considering the above factors, if the operation wavelength range transfer to the longer wavelength, the silica absorption loss is an interference factor and the corresponding equipment may be more expensive. If the operation wavelength range transfer to the shorter wavelength, it is more difficult to achieve higher sensitivity because of the weak coupling between the core mode and SPP mode.
The full-width-half-maximum of each confinement loss spectrum caused by the SPR effect is only about 20 nm, which means a sharp spectrum, while the absorption peaks may be wider than that of the SPR-induced loss spectra, so it can be identified clearly. Under normal circumstances, the absorption peaks of many substances are fixed and known, while the RI of the analyte is also related to the concentration. Thus, if the type of analyte is known, while the RI of the analyte at the corresponding concentration is not known, we can use the loss peak caused by the SPR effect to measure the refractive index of the analyte.
3) As claimed in the Conclusion section, this type of optical sensor could be used for application in the biochemistry and medicine. Authors had better offer a couple of lines on how the biochemical surface treatment required for biochemistry sensing can be performed practically, for example, on which surface of this fiber structure bio-recognition elements such as antibody or aptamer, or DNA can be immobilized.
Authors’ response: We thank the reviewer for his/her comments. Biochemistry and medicine include the bio-recognition, anti-body or DNA testing, etc. As a matter of fact, the RIs of many halogenated ethers and pharmaceuticals are located in the lower RI range. For example, sevoflurane has a RI of ~1.27 and serves as the anesthetic in the medical field [1-2]. Moreover, in water pollution or aquatic environmental measurements, the RIs of a lot of analytes are also located in the low RI range [3]. Our designed sensors mainly measure the RIs of liquid chemicals. However, in bio-sensing, the RIs of some analytes with bio-molecules are not belonging to the low RI range. In some cases, helping agents are required to keep the bio-substance under investigation in the sensing field of the biosensor. For instance, the Poly-L-lysine (PLL) is commonly used to immobilize biomolecules such as deoxynucleic acid (DNA), and it will immobilize itself to a solid surface. Hence, the molecular interactions of DNA can be detected by a PLL bound surface. In general, the PLL and DNA layers have a refractive index range of 1.45 to 1.48 [4].
[1] Chen X, Xia L , Li C . Surface plasmon resonance sensor based on a novel D-shaped Photonic Crystal Fiber for low refractive index detection. IEEE Photonics J, 2018, 10, 1-9.
[2] Dean J.A., Lange’S Chemistry Handbook Version 15th, McGraw-Hill Book Company, 1998, Section.1
[3] Govindan G., Raj S.G., and D. Sastikumar, “Measurement of refractive index of liquids using fiber optic displacement sensors,” J Amer. Sci. 2009, 5, 13–17.
[4] Chu S, Kaliyaperumal. N , Abobaker A M , et al. Design and Analysis of Surface Plasmon Resonance based Photonic Quasi-Crystal Fiber Biosensor for High Refractive Index Liquid Analytes. IEEE J Sel Top Quant, 2018, 25:1-9.

Round 2
Reviewer 2 Report
On the whole, proper Revisions have seemed to be made to get it publishable except for some points listed below
1. The reference added at [11] does not support the representative prototype of the plasmonic RI sensors because it was greatly aided by graphene. Authors need to choose the literature that better suits the pupropse of the context of the introduction, for example, the SPR device as a RI sensor that appeared for the first time, and those that reported the best resolution in RI sensing, and etc.
2. Some more minor amendments such as spell check and English grammars appear to be needed.
Author Response
Manuscript Number: Sensors-994510
Title: Hollow-Core Negative Curvature Fiber with High Birefringence for Low Refractive Index Sensing Based on Surface Plasmon Resonance Effect
Authors: Shi Qiu, Jinhui Yuan*, Xian Zhou, Feng Li, Qiwei Wang, Yuwei Qu, Binbin Yan, Qiang Wu*, Kuiru Wang, Xinzhu Sang, Keping Long, And Chongxiu Yu.
Affiliations:
State Key Laboratory of Information Photonics and Optical Communications, Beijing University of Posts and Telecommunications, Beijing 100876, China
Research Center for Convergence Networks and Ubiquitous Services, University of Science & Technology Beijing, Beijing 100083, China
Photonics Research Centre, Department of Electronic and Information Engineering, The Hong Kong Polytechnic University, Hung Hom, Hong Kong
Department of Physics and Electrical Engineering, Northumbria University, Newcastle upon Tyne, NE1 8ST, United Kingdom
Dear Editor,
Thank you very much for your decision about accepting this work after minor revisions. We received the reviewers’ comments and would like to thank them for their pertinent and valuable feedback. We made all the necessary changes in the manuscript according to the reviewers’ comments. Please find following detailed descriptions of changes where they are performed. We have copied the reviewers’ comments before responding to it to facilitate the easy reading of this document. The revised parts in the original manuscript are marked in red font.
Please let us know if additional information is necessary to support our manuscript.
Best regards,
Shi Qiu, on behalf of the other authors.
Reviewers' comments:
Reviewer #2: On the whole, proper Revisions have seemed to be made to get it publishable except for some points listed below
1) The reference added at [11] does not support the representative prototype of the plasmonic RI sensors because it was greatly aided by graphene. Authors need to choose the literature that better suits the pupropse of the context of the introduction, for example, the SPR device as a RI sensor that appeared for the first time, and those that reported the best resolution in RI sensing, and etc.
Authors’ response: We thank the reviewer for his/her comments. We have deleted the description on the former reference and added the description on the first reported work on the SPR refractive index sensor. This work also experimental demonstrated that the gold based SPR fiber sensor can be used as a RI sensor. The revised content is as following
“In 1999, Slavı´k et al firstly proposed a gold-based SPR refractive index (RI) sensor, whose sensitivity and resolution can achieve 3300 nm/RIU and 3×10-5 RIU in the RI range of 1.324 to 1.335, respectively [11].”
“[11] Slavı´k R, Homola J ,Čtyroký J. Single-mode optical fiber surface plasmon resonance sensor. Sensors & Actuators B Chemical 1999, 54, 74-79.”
2) Some more minor amendments such as spell check and English grammars appear to be needed.
Authors’ response: We thank the reviewer for his/her comments. We have already checked and corrected the English grammar and spelling of the whole paper.
